# The Synthesis and Evaluation of RGD−Conjugated Chitosan Gel as Daily Supplement for Body Weight Control

**DOI:** 10.3390/ma14164467

**Published:** 2021-08-10

**Authors:** Wei-Yao Chen, Yu-Ting Chen, Cherng-Jyh Ke, Ching-Yun Chen, Feng-Huei Lin

**Affiliations:** 1Institute of Biotechnology, National Taiwan University, Taipei 106216, Taiwan; tp6jo3ul6@hotmail.com; 2Institute of Biomedical Engineering, National Taiwan University, Taipei 106216, Taiwan; tingamy93@gmail.com; 3Biomaterials Translational Research Center, China Medical University Hospital, Taichung 40202, Taiwan; fonchanwd@gmail.com; 4Department of Biomedical Sciences & Engineering, National Central University, Taoyuan 32001, Taiwan; chingyun523@gmail.com; 5Institute of Biomedical Engineering and Nanomedicine, National Health Research Institutes, Miaoli County 35053, Taiwan

**Keywords:** obesity, RGD, chitosan, type 2 diabetes, highly biocompatibility

## Abstract

(1) Background: Obesity is one of the most widespread chronic diseases and increases the risk of several other chronic diseases, especially type 2 diabetes. (2) Methods: Endobarrier is a new medical device what is worn in the small intestines for the treatment of type 2 diabetes and obesity. However, given the invasive and other adverse effects of the Endobarrier, we propose the use of RGD peptide conjugated with chitosan (RC) as an alternative. (3) Results: The FTIR and NMR spectrum showed RGD peptide was successfully conjugated on chitosan and RGD−CT is retained in the small intestine even after digestion. In vitro of wst-1 and live and dead staining studies show that the RGD−CT gel is highly biocompatible and non-toxic. Rats treated with the RGD−CT gel for a short term showed significant decrease change more than 30% in body weight, while the blood and hematic biometrics were within normal values. (4) Conclusions: The RGD−CT gel is safe, suitable for the short-term, reducing visceral fat rate health food to control weight. In the future, it is expected to develop a safe, long-term effective, flexibility of use and low-side-effect anti-obesity therapy in the era of precision medicine by further modification.

## 1. Introduction

Obesity arises from the energy imbalance between food intake and calorie expenditure. According to the WHO, the body mass index (BMI) which is calculated based on an individual’s body weight and height is the easiest and the most common way to categorize an individual as underweight, normal weight, overweight, or obese [1]. A BMI over 25 kg/m^2^ falls within the overweight and obesity range and it is estimated that 60% of the population worldwide will be obese by 2030. Considering the potential advances in developing countries, this is potentially an underestimation [2,3]. In fact, obesity is one of the most widespread chronic diseases around the world and it elevates the risk of several other chronic obesity-related diseases, especially type 2 diabetes [1,4].

Lifestyle interventions focusing on dietary control and exercise plan are common ways to regulate body weight, but they have limited effects and low efficiency. Previous studies have indicated that due to the body’s adaptation to physiological neurohormonal changes, reduction in food intake will eventually lead to weight regain [5].

Advances in pharmacotherapy have provided other options to treat obesity. To date there are only five types of medications approved by the US Food and Drug Administration (FDA) for long-term treatment of obesity. These include the lipase inhibitors (Roche, Basel, Switzerland, Orlistat, Xenical^®^), serotonin agonists (Arena Pharmaceuticals, San Diego, CA, USA, Lorcaserin, Belviq^®^), combination of phentermine-opiramate (VIVUS Inc., Campbell, CA, USA, Qsymia^®^), glucagon-like peptide1 liraglutide (Novo Nordisk, Denmark, Saxenda^®^) and a combination of bupropion-naltrexone (Orexigen, CA, USA, Contrave^®^) [6]. However, there are potential hazards associated with these all these pharmaceutical treatments. Sibutramine (Abbott Laboratories, Chicago, IL, USA, Reductil^®^), another anti-obesity treatment reduces appetite by inhibiting the central nerve re-uptaking norepinephrine, serotonin, and dopamine. Its side effects include hypertension and heart disease. It was launched in 1998 but was withdrawn in 2010 [7].

The small intestine plays an important role in the digestive absorption and control of blood glucose levels which are regulated by endocrine responses and glucose intake. Endobarrier is a medical device which is worn inside the body for the treatment of type 2 diabetes and obesity [8]. It is a thin plastic sleeve that is implanted by an endoscopic bariatric procedure which prevents the body from digesting food within the upper part of the small intestine. Disruption of the digestive absorption in the upper part of the small intestine leads to changes in fatty acid and glucose metabolism through a variety of mechanisms, including modulation of gut hormones, alterations in the gut bacteria and disruption of bile flow. The Endobarrier has two operators that require both an implant and an explant. Prior to the implant, the patients are required to take protein pump inhibitors and continue them throught the implant period and for some time after the device is removed. The implant is then delivered endoscopically under fluoroscopic X-ray guidance, which exposes the patients to radiation. Patients have to be on a liquid diet during the EndoBarrier placement and for a short duration after the procedure [8,9].

Given these inconveniences associated with the Endobarrier usage, search for alternative options has been on. Chitosan (CT) is a new substance that can potentially replace the EndoBarrier. CT is a biodegradable, biocompatible, and bioplastic material which is very safe for use in health and biotechnology products such as drug carriers and scaffolds [10,11]. It is a weak base with a pKa of 6.5, and can therefore, bind to free fatty acids and bile acids. As ionically binding anions [12], the coacervation of CT’s positively charged amino groups and the negatively charged fatty acids and bile acids results in the formation of mixed micelles [13,14]. As the pH gradually increases in the proximal small intestine, the micelles are disrupted by the removal of the bile acids or fatty acids. Furthermore, in the neutral pH environment, bile acids bind with CT to a greater degree compared to fatty acids. In other words, CT, which is a health supplement, could bind to free fatty acids efficiently in the stomach but not in the small intestine [15]. This adhesive capability of CT can be further enhanced by conjugation of suitable targeting ligands, such as peptides that specifically bind to the intestinal epithelial cells even in a neutral or alkaline pH environment [16,17].

The RGD peptide is made up of arginine, glycine, and aspartic acid. It has been identified as the minimal recognition sequence within integrins such as α5β1, αVβ1, and α8β1 required for cell attachment. The RGD peptide has been widely used in nanoparticles to target tumor cells, and for coating implantable medical devices to avoid immunological rejection. RGD−conjugated CT (RC) gel could be developed as an oral health supplement for anti-obesity treatment, based on its biocompatibility, low risk, and ability to coat the surface of gastrointestinal tract for a medium term, to form an absorption barrier. The gel is designed to be similar to the EndoBarrier^®^, taking advantage of the affinity of the RGD peptide for integrins.

The RGD−CT was prepared and then checked using Fourier-transform infrared spectroscopy (FTIR), Nuclear Magnetic Resonance (NMR), and Ninhydrin test to confirm functional groups, molecular structure, respectively. We hypothesize that the RGD−CT will be effective in regulating body weight. We focus on the results of the following: cell viability and cytotoxicity on a cellular level, chronic toxicity, body weight control, blood analysis, serological analysis, and sectioning examination of internal organs. In this study, we describe the conjugation of the RGD peptide to CT and evaluate the chemical properties of the conjugated product and its therapeutic effects as a supplement for body weight control both in in vivo and in vitro models.

## 2. Materials and Methods

### 2.1. Preparation of the RGD−CT Gel

Fully deacetylated chitosan (Carbosynth Ltd., Compton, UK, YC06764, 190–310 kDa) was dissolved in 1M acetic acid at 2% *w*/*v* concentration and 3000 rpm for 30 min centrifuged to remove impurities in the solution. Next, 0.45 mol 1, 4-butanediol diglycidyl ether (BDDE) (Sigma-Aldrich Ltd., Burlington, MA, USA, 220892), 10% *v*/*v* of 100 mL isopropanol, and 3 mg RGDs (NHRI)peptide powder were added and stirred to allow cross-linking for 3 h at room temperature. The solvent was then removed, freeze-dried, and stored at −80 °C. Figure 1 shows the final product.

### 2.2. Fourier-Transform Infrared (FTIR) Spectrometry

A Fourier-transform infrared spectrometer (PerkinElmer-FTIR spectrum100) was used to identify the functional groups of organic and polymers compounds by measuring absorption of infrared radiation of wavelengths. It was used to identify qualitative changes in the functional groups after conjugation. After correcting for the blank background, dry samples prepared with CT, RGD−CT and RGD powder were placed on the holder and scanned. The scan range was set to a 400–4000 cm^−1^ wavelength and 4 cm^−1^ resolution, and 16 scans were obtained.

### 2.3. Nuclear Magnetic Resonance (NMR) Spectroscopy

The nuclear magnetic resonance (NMR) spectroscopy is based on the spin and magnetic moments of the nucleus; therefore, in a strong magnetic field, it will induce energy-level splitting and can be used to analyze the molecule structure. Samples (10 mg) in 0.5 mL of 10% acetic acid-D_2_O (Fluka, St. Gallen, Switzerland) were taken in 5 mm NMR tubes and analyzed in an NMR spectrometer (Bruker, Billerica, MA, USA, Bruker Avance III 600 MHz). The one-dimensional NMR proton spectrums (^1^H) for CT and RGD−CT were compared to determine if the RGD immobilization was successful.

### 2.4. RGD Functionality Test

To evaluate the RGD functionality in digestive fluids, we set up an in vitro gastric and intestinal environment according to the method described by Knorr et al. [18]. We added 100 mg of CT and RGD−CT to 10 mL of 0.1 M hydrochloric acid solution (pH: 1.0–2.0) and stirred in a 37 °C water bath for 2 h. We then added 0.1M sodium hydroxide solution until the pH increased to 6.5–7.5. The samples were then centrifuged to remove the additional solution and freeze dried.

### 2.5. Ninhydrin Test

The ninhydrin was obtained from Sigma-Aldrich N4876. A chemical test, ninhydrin test, is a way to check whether a given analyte contains amines or α-amino acids and Free amino groups will react with the reagent to yield a purple solution. Dissolving Ninhydrin powder in 95% alcohol solution at 3% *w*/*v*, then adding 2M acetic acid buffer in regents with 50% volume ratio, and the regents stored in dark at 4 °C. Standard samples were prepared by adding 1.2, 1, 0.8, 0.6, 0.4, 0.2 and 0 mL of 0.1 mmol cysteine solution. RGD−CT was dialyzed in Ninhydrin solution, and we calculated the amount of conjugated RGD by measuring the amount of unconjugated amino acid. the 2, 2-dihydroxyindane-1, 3-dione (Ninhydrin) was used to examine the amount of amine groups in samples and the color of the solution changed from light yellow to purple, which could be quantified at 570 nm by spectrophotometer microplate reader.

### 2.6. WST-1 Assay for Cell Proliferation

WST-1 kit was obtained from Roche, Inc. (Basel, Switzerland) 05015944001 and used to evaluate cell viability. IEC-6 (American Type Culture Collection, Manassas, VA, USA, ATCC Number CRL-1592), an intestinal epithelial cell line was cultured in high glucose Dulbecco’s modified eagle medium supplemented with 5% fetal bovine serum, 1% antibiotics and 2 c.c/L insulin (insulin transferrin selenium mixture, ITS-M) at 37 °C in a 5% CO_2_ incubator. After the cells stabilized in culture, they were seeded into 96-well plates at 3 × 10^3^ cells/well for biocompatibility tests. The next day, cells were washed with phosphate-buffered saline (PBS), and the extractive solution (200 μL per well) of CT and RGD−CT were added for 3 days, followed by ISO-10993 standards; the extractive solution from co-incubating chitosan or RGD−chitosan with medium for 24 h. Cell viability was measured using a tetrazolium salt (WST-1), which is cleaved to produce a soluble formazan dye by the action of succinate-tetrazolium reductase in the mitochondrial respiratory chain of the viable cells. Optical density (OD) was measured the absorbance on a spectrophotometer microplate reader with a test wavelength at 450 nm and a reference wavelength at 630 nm and the percent cell viability was calculated using the following equation:(1)$$\mathrm{Cell}\;\mathrm{viability}\;(\%)=\frac{\mathrm{Experimental}\;\mathrm{value}}{\;\mathrm{control}\;\mathrm{value}}\times 100\%$$

### 2.7. Live and Dead Cells Staining

We used the LIVE/DEAD^®^ Viability/Cytotoxicity Kit for mammalian cells (Invitrogen™) (Thermo Fisher Scientific, Waltham, MA, USA, L3224) to visualize the live and dead cells. After seeding IEC-6 cells into 12-well plates at 5 × 10^4^ cells/well, we added 1 mL of the material solution (CT/RC) to each well and cultured for them 4 h. At the end of this treatment, we removed the medium, washed the cells with PBS, and then added 5 µL/mL of Calcein AM and 1 µL/mL of propidium iodide (PI) to each well. The reagents were allowed to react for 20 min in the dark and were finally washed with PBS and observed under a fluorescence microscope.

### 2.8. Dose Optimization

To evaluate the optimal effects of CT and RGD−CT on cell viability, CT and RC solutions (1, 2, and 3 mg/mL prepared in 0.01M acetic acid) were tested in IEC-6 cells for one and three days, respectively, using the WST-1 cell viability assay.

### 2.9. Cell Adhesion Test

To test whether the RGD−CT promotes IEC-6 cell adhesion, we used ibidi micro fluid channel seeded IEC-6 cells to simulate the intestinal environment. First, the cell suspension (1 × 10^6^ cells/mL) was injected into the device (length: 50 mm, width: 5 mm, height: 800 mm) and incubated for 24 h. At the end of the incubation, we removed the medium and added Hoechst and Safranin O dyes at a concentration of 1 μL/mL to stain the nucleus and cell-secreted mucins. CT and RGD−CT containing medium were passed through the channel at a flow rate of 750 μL/h using a syringe pump according to a previously described protocol [19,20]. After allowing the solutions to flow for 30 min to mimic a meal, the channel was stored in glycerol and observed under a fluorescence microscope.

### 2.10. Animal Model

Male Sprague Dawley rats (10 weeks old) were obtained from Biolasco. This ethical code number of animal experiment is NHRI-IACUC-106159-A and this protocol has reviewed and approved by the Institutional Animal Care and Use Committee (IACUC). The rats were housed under a 12 h light/12 h dark cycle and controlled for temperature and humidity. The animals had free access to food and water and were studied after 1 week of adaptation to the lighting conditions. The rats were treated according to the guidelines of the Institutional Animal Care and Use Committee, National Health Research Institutes. At the beginning of the trial, 6 rats with the closest body weights were randomly assigned to the same group. The animals were divided into three groups (6 rats in each) and were given the following diets: normal control (NC), CT by gavage, and RGD−CT by gavage. The CT and RGD−CT gel solutions were both given at 30 wt.%/100 g of body weight according to the US FDA dose conversion method. The NC group was given saline based on the same schedule. Each rat was gavaged three times a week for 6 weeks to evaluate the efficacy of the RGD−CT gel.

### 2.11. Change in Body Weight

The changes in bodyweight and standard weight gain were calculated using the following equations:Percent Weight change = (Weight_final_ − Weight_original_)/Weight_original_ × 100%(2)

### 2.12. Serum Chemistry and Hematology

After 12 h of fasting, the rats were deeply anesthetized with isoflurane, and blood was drawn from the heart, before sacrificing them. Blood was centrifuged at 3000 rpm for 10 min. The supernatants were collected and used for studying the serum chemistry. Blood mixed uniformly with EDTA was used for hematological analysis by flow cytometry.

### 2.13. Percent Body Fat

We collected adipose tissue from the visceral fat (epididymal, mesentery, perirenal, retroperitoneal adipose tissue) and subcutaneous fat (anterior, posterior, dorsolumbar, inguinal, gluteal adipose tissue).

The percent body fat was calculated as follows:Body fat (%) = [Total visceral fat weight (g)/Body weight (g)] × 100%(3)

### 2.14. Statistical Methods

Data were expressed as mean ± standard deviation of at least three replicates. Statistical analyses were performed by one-way ANOVA and t-test analysis of the variance test. The results were considered significant when the *p*-value was <0.05.

## 3. Results

### 3.1. Material Analysis

As shown in Figure 2 the RGD peptide and RGD−CT have absorption bands of 1735–1750 cm^−1^ and 1640–1690 cm^−1^, respectively, while that for CT has different absorption bands of 1400 cm^−1^. In addition, compared to RGD and RC, both RGD−CT and CT showed the peak of 1000–1300 cm^−1^, indicating the presence of ether bonds, which provide the nucleophilic attack to the epoxy structure that caused the conjugation. Besides, the CT polymer itself had an ether bond. Based on the above results, it was assumed that the RGD peptide had successfully conjugated with CT via BDDE [21]. As Appendix A shows, the diluted cysteine solution was measured for standard curve of free amino group concentration as shown in Appendix A (R^2^ = 0.99). The summation of free amine concentration from five dialysis samples was 6.72 × 10^−6^ (mol/g) and 1.67 × 10^−7^ (mol/g); however, considering there were two amine groups in each RGD structure, the final concentration of conjugated RGD was 3.56 × 10^−8^ (mol/g). The results were greater than 0.6 pmol, which the minimum RGD concentration for cell adhesion.

The NMR proton spectrum in Figure 3 shows the additional chemical shifts (δ) in RGD−CT compared to CT at 1.2806 ppm and 8.2292 ppm. The chemical shift at 1.2806 ppm was due to the formation of a carboxy long chain during conjugation, and that at 8.2292 ppm was due to an additional indole amine, which arginine and glycine possessed. Since the CT was nearly 90% deacetylated, it can be inferred that the source of the amide bonds was the RGD peptide. Therefore, based on the ^1^H NMR, it could be concluded that the RGD peptides had conjugated with CT, resulting in molecular changes [22].

To determine whether the RGD short peptides would be inactivated by digestive fluids, the RGD−CT was allowed to react with simulated gastric and intestinal fluids in vitro (defined as After-RC), and its FTIR spectrum was compared with the original RC. As shown in Figure 4A, the characteristic absorption peaks for RGD were 1750, 1680, and 1400 cm^−1^. A comparison between RGD−CT and after-RC showed that the peaks were retained although at a slightly lower intensity, which was attributed to the decomposition in the in vitro digestive fluid environment. These results show that the RGD peptide on the RGD−CT retained its functional group even after digestion [23]. Figure 4B–G shows fluorescent staining overlay for Control group, CT group and RGD−CT group at 40 min and 12 h. At 12 h, small fragments of the materials are left.

### 3.2. In Vitro Studies

Figure 5B shows that cells treated with 3 mg/mL of RGD−CT in 0.01M acetic acid solution had the highest viability based on a 3-day assay. The viability of CT and RGD−CT treated cells were comparable to those of control cells. The morphology of these cells (Figure 5A) visualized by optical microscopy showed no changes, thereby indicating that RGD−CT is highly biocompatible. Additionally, live cell staining following CT and RGD−CT treatments was comparable to that in the control cells (Appendix A). The poor PI staining in combination with the WST-1 data confirms that the RGD−CT is not toxic in vitro.

### 3.3. In Vivo Studies

Figure 6 shows the change in the average body weights in the NC, CT, and RGD−CT groups. A significant difference lower (*p* < 0.05) was seen between the NC and RGD−CT groups after 4 weeks of treatment, while no significant difference was seen between the NC (414.6 ± 5.9 g) and CT (401.9 ± 3.2 g) groups. The percent weight change in 4 weeks was significantly lower in the RGD−CT (384.0 ± 4.2 g) group when compared to the NC group. However, this difference became less obvious after 4 weeks. Compared to the NC group, the RGD−CT group had a 62% weight gain rate in 4 weeks, which increased to 70% after 4 weeks. These results suggest that the conjugated CT could regulate weight more effectively than its non-modified version. Besides, the RGD−CT gel appeared to be more effective when used for short-term (between 4 to 6 weeks), which was consistent with the findings of previous clinical trials [24,25].

## 4. Discussion

Previous studies and data from clinical trials of the Endobarrier have indicated device-related adverse effects during the implant periods, which often require premature explant for melaena and device migration resulting in the blockage of the duodenal-jejunal bypass liner with food and abdominal pain [8]. The explant of the Endobarrier between 12 to 26 weeks, endoscopic surgery is the only way to deal with the discomfort during the treatment. Despite its ease of performance compared to traditional surgery, endoscopic surgery is still inconvenient for patients. RGD−CT gel is, therefore, a good alternative to endoscopic surgery.

The viability and morphology of CT and RGD−CT treated cells (Figure 4 and Appendix A) shows that RGD−CT is highly biocompatible and not toxic. It demonstrates that RGD peptide and chitosan modified by BDDE do not causes increased toxicity. Our in vivo results show that 4–6 weeks of continuous use of the RGD−CT gel resulted in a significant difference lower in body weight with no side effects, confirming our hypothesis. The blood and hematic biometrics were within the normal values (Appendix A) and were consistent with those reported previously [26]. Besides, the RGD−CT gel can be administered orally, has no side-effects and requires a brief stay in the body.

## 5. Conclusion

This study developed a method to combat obesity and diabetes through a safe and without negative side-effect oral intake of RGD−modified chitosan hydrogel.

It does not cause any inconvenience to the patients, and in any cases of any discomfort, its administration can be stopped immediately. Moreover, considering the high cost of RGD, patients can adjust the frequency of the oral intake of the RGD−CT gel as per their needs. Given the flexibility of use and low-side-effect RGD−CT gel is a new anti-obesity therapy in the era of precision medicine [26].

## Figures and Tables

**Figure 1 materials-14-04467-f001:**
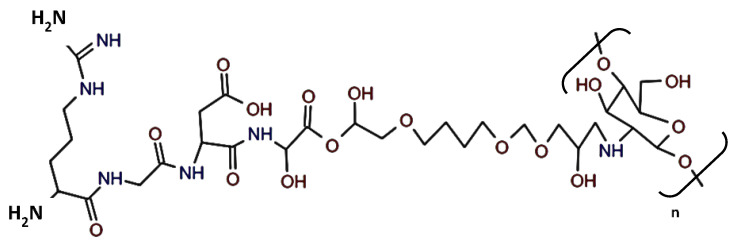
Chemical structure of the RGD−Chitosan conjugate.

**Figure 2 materials-14-04467-f002:**
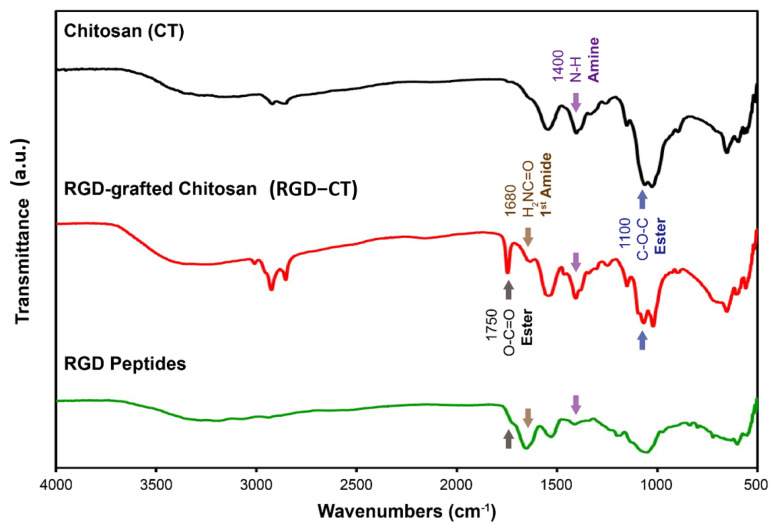
FTIR analysis of the RGD peptide, RGD−Chitosan conjugate, and fully deacetylated chitosan. The stretching vibration absorption band of the ester group (OC=O) is at 1750 cm^−1^, while that for the primary amine bond (H2NC=O) is at 1680 cm^−1^, and ester bond is at 1100 cm^−1^.

**Figure 3 materials-14-04467-f003:**
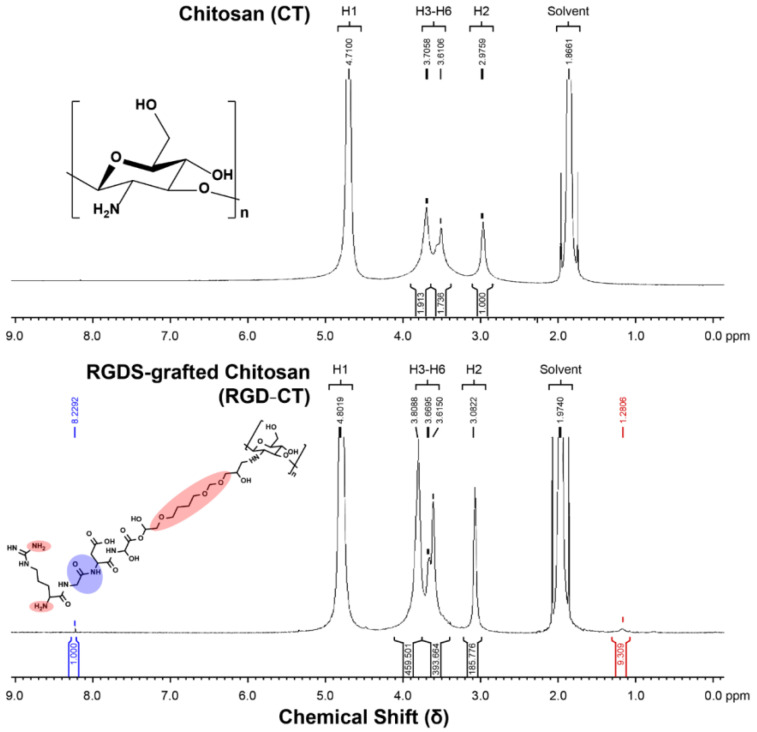
The NMR analysis of 90% deacetylated chitosan and the RGD−Chitosan conjugate.

**Figure 4 materials-14-04467-f004:**
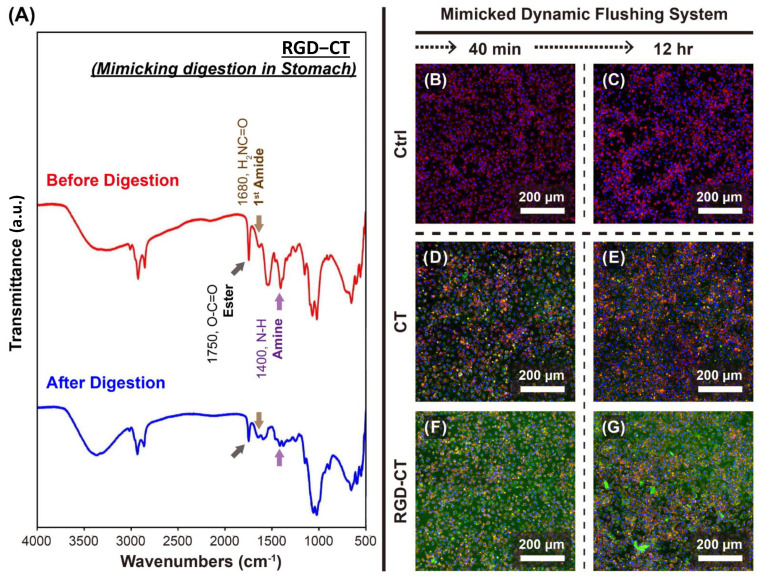
In vitro simulated cell adhesion test. Cell adhesion abilities of chitosan (CT) and RGD−Chitosan conjugate (RC) were tested as described in the methods. (**A**) FTIR analysis of RGT-CT after mimicking digestion. Shown here is the fluorescent staining overlay for (**B**) Control group 40 min, (**C**) Control group (**D**) CT group 40 min, (**E**) CT group 12 h, (**F**) RGD−CT group 40 min, and (**G**) RGD−CT group 12 h. The RGD−CT group shows more material residue compared to the CT group. At 12 h, small fragments of the materials are left.

**Figure 5 materials-14-04467-f005:**
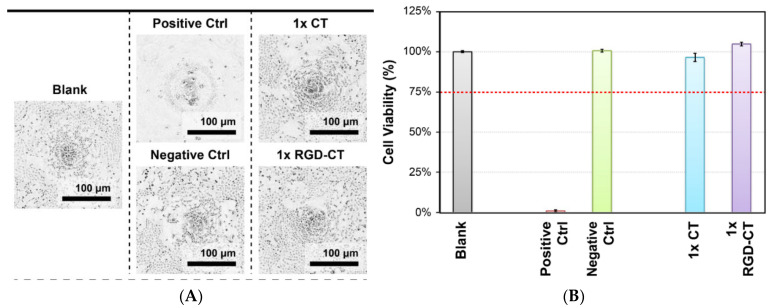
Cell viability tests. (**A**) EC-6 cells morphology of Blank, Positive control, Negative control, Chitosan (1 × CT), and RGD−Chitosan conjugate (1 × RGD−CT). (**B**) WST-1 test of EC-6 cells in of Blank, Positive control, Negative control, Chitosan (1 × CT), and RGD−Chitosan conjugate (1 × RGD−CT) for three days. 1 × CT and 1 × RGD−CT is 3 mg/mL.

**Figure 6 materials-14-04467-f006:**
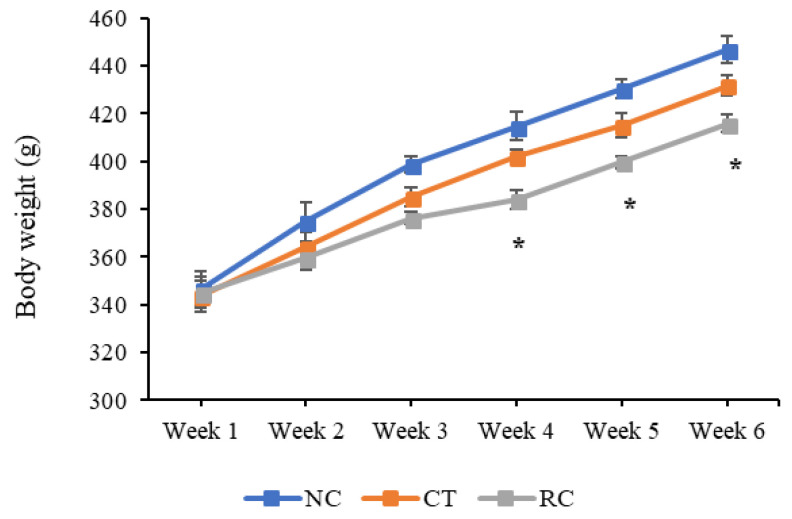
Change in average body weights. Shown are the changes in body weights in the normal control (NC), chitosan (CT) and RGD−Chitosan conjugate (RC) groups. The NC and the RGD−CT groups show significant changes after 4 weeks of treatment (*: *p* < 0.05, *n* = 6).

## Data Availability

No new data were created or analyzed in this study. Data sharing is not applicable to this article.

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
