# Peer review of "The Synthesis and Evaluation of RGD−Conjugated Chitosan Gel as Daily Supplement for Body Weight Control"

_materials, 2021, doi:10.3390/ma14164467_

Round 1
Reviewer 1 Report
The second version of materials-1315469 (which was previously polymers-1262568) became much more qualitative and can be accepted for publication.
Author Response
Thank you so much! Please keep safe and stay away from the pandemic Coivd-19. with best regards
Reviewer 2 Report
The authors have done all the suggested revisions thus it can be proposed for publication.
Author Response

(The authors gave the same response as above.)

Reviewer 3 Report
The authors of the paper entitled “The Synthesis and Evaluation of RGD-Conjugated Chitosan Gel as 3 Daily Supplement for Body Weight Control” have taken a very comprehensive approach to the subject. The entire pathway from synthesis to preclinical studies is presented here. I am impressed by the amount of measurements and results presented and I really like this holistic approach.
I think the work is worth publishing but a few imperfections should be fixed.
In my version of the document, some of the text is in black, some in red. It should be unified
Linie 145 The author writes about „Optical density (OD) was measured at 450 nm” but he was certainly measuring absorbance which is proportional to concentration . Optical density is identified with the scattering of the light and the counts of bacteria.
Figure 2 With such transposed graphs (one on top of the other) transmittance can no longer be expressed as %, the unit should be relative [a.u]. The analogous case is in Figure 5.
Linie 97 alutors write about „Fully deacetylated chitosan” while in the description to the graph figure 3 it is written „The NMR analysis of 90% deacetylated chitosam” What is the degree of chitosan deacetylation the authors are working with. There is also no information about its molecular weight in the materials section. These two parameters translate into biological properties and need to be clearly specified.
Linie 198 Determination of „Percent body fat” should be described in more detail
Author Response
The old manuscript has been fully re-written by major revision based on the previous suggestions/comments from "Polymers". In addition, the pictures and photos were also re-arrangement in the series from reviewers' suggestions. we are already unified in the last version manuscript.
Thank you so much! Please keep safe and stay away from the pandemic Coivd-19. with best regards

Round 2
Reviewer 3 Report
The authors have improved the work as suggested in a satisfactory manner. I would like to thank them for the work done. In this form, the work can be published.
This manuscript is a resubmission of an earlier submission. The following is a list of the peer review reports and author responses from that submission.
Round 1
Reviewer 1 Report
Chitosan is not a new substance as the author’s state.
In Material and methods specify the Molecular weight of the chitosan used and also its source?
Information regarding the source of cell line IEC-6, an intestinal epithelial cell line, is missing.
“ELISA spectrophotometer” must be replaced by “spectrophotometer microplate reader “
In section “2.6. WST-1 assay for cell proliferation” is missing the time of exposure of cells to extracts. Moreover is not clear how to do extract of chitosan solution and also modified chitosan?
In section “2.10. Animal mode” is missing the respective authorization of the animal experimentation by the local authorities.
The discussion of the results lacks comparison with other studies in which chitosan was used as a weight control. The mechanism of chitosan in weight control is not addressed, nor is the advantage of peptide modified chitosan and what the implications are for weight control is understood.
Author Response
We already answer the questions and the modified parts are marked as red.
If there is still room for improvement, please let me know.
Reviewer 2 Report
The manuscript polymers-1262568 demonstrates the development of methods of obtaining chitosan derivatives, study of their properties and in my opinion can be published in Polymers journal.
However, I think that the manuscript can be improved for reading. This will increase interest for a wider range of readers. Here are some suggestions:
- Abstract should show the essence of the article, in my opinion, and not be only a summary of what the authors did.It is not clear what the achievements of the authors are compared with the known data. What is fundamentally decided, what effect is achieved.Abstractmustberewritten.
- Lines 99–102: It is necessary to provide a detailed procedure for the synthesis of the derivative.
- Figure 1: It is necessary to give a scheme of the reaction, indicate on it the degree of functionalization of the BDDE and RGD. Incorrect product formula: part of the formula is not shown, there are extra atoms.
- Lines 115–116: What acetic acid was used to record the NMR spectra?
- Lines 212–218: The discussion of Figure 2B, as well as the figure itself, is not very informative for the reader. I propose to transfer this material to Supplementary. It is not clear what degree of modification of chitosan with the RGD peptide has been achieved.
- Lines 219–222: The discussion and interpretation of the NMR spectrum (Figure 3A) is incorrect. The spectra themselves do not allow the reader to reveal either the chemical shift or the integral intensity. It is necessary to bring the spectra separately, for example, in Supplementary or make them readable.
- How is Figure 3B different from Figure 2A?
- Figure 4: Poorly readable image format. It is necessary to correct the form of presentation of the material. What does the fourth column from the left refer to? It is not indicated in the legend.
- Incorrect conclusions without evaluating the effects of RGD peptide and chitosan modified by BDDE.
- An extended explanation of the findings needs to be added (Figure 6).
- It is necessary to add a conclusion.
Author Response
Dear reviewer
We already answer the questions and the modified parts are marked as red.
If there is still room for improvement, please let me know.
Round 2
Reviewer 1 Report
The authors still do not improve the article as suggested in my last review. In particular in the following aspects:
In section “2.6. WST-1 assay for cell proliferation” is missing the time of exposure of cells to extracts. Moreover is not clear how to do extract of chitosan solution and also modified chitosan?
The discussion of the results lacks comparison with other studies in which chitosan was used as a weight control. The mechanism of chitosan in weight control is not addressed, nor is the advantage of peptide modified chitosan and what the implications are for weight control is understood.